# Freedive Training Gives Additional Physiological Effect Compared to Pulmonary Rehabilitation in COPD

**DOI:** 10.3390/ijerph191811549

**Published:** 2022-09-14

**Authors:** Zoltán Csizmadia, Pongrác Ács, Gergő József Szőllősi, Blanka Tóth, Mária Kerti, Antal Kovács, János Tamás Varga

**Affiliations:** 1Faculty of Health Sciences, University of Pécs, 7621 Pécs, Hungary; 2Faculty of Health Sciences, University of Debrecen, 4032 Debrecen, Hungary; 3Department of Pulmonary Rehabilitation, National Koranyi Institute of Pulmonology, 1122 Budapest, Hungary; 4Department of Pulmonology, Semmelweis University, 1083 Budapest, Hungary

**Keywords:** breath holding, respiratory rate, respiratory training, respiratory effectiveness, pulmonary rehabilitation

## Abstract

Introduction: Pulmonary rehabilitation (PR) is beneficial for lung mechanics, chest kinematics, metabolism, and inspiratory and peripheral muscle function. Freediving training (FD) can be effective in sportsmen and can improve breath-holding time. Aims: We sought to determine the effectiveness of freediving training in the pulmonary rehabilitation of COPD patients. Patients and methods: Twenty-three COPD patients (15 men and 8 women; median age 63 years; FEV1: 41% pred; BMI: 28 kg/m^2^) participated in the FD + PR group (3 weeks PR and 3 weeks FD + PR) and 46 patients with COPD (25 men and 21 women; median age 66 years; FEV1: 43% pred; BMI: 27 kg/m^2^) participated in an inpatient PR program (6 weeks). Patients performed comfort zone breath holding for 30 min/day. Patients increased their breath-holding time within their comfort zone for 30 min. We detected lung function, chest expansion (CWE), inspiratory muscle pressure (MIP), peripheral muscle function (GS), and exercise capacity (6MWD), and we included breath-holding time (BHT), quality of life score (COPD Assessment Test (CAT)), modified Medical Research Dyspnea Scale (mMRC) score, and the severity of the disease assessed by the BODE index (FEV1, BMI, 6MWD, and mMRC) and an alternative scale (FEV1, BMI, 6MWD, and CAT). Result: There were significant differences in the characteristics of the two groups. Significant improvement was detected in all functional and quality of life parameters except lung function in both groups. Significantly higher improvement was detected in CWE, GS, 6MWD, BHT, CAT, mMRC, alternative scale, and MIP. The improvement in forced vital capacity (FVC) was not significant. There were no side effects of FD training. Conclusion: The FD method can potentiate the effect of PR, improving not only BHT but also other parameters. Trial registration: ISRCTN ISRCTN13019180. Registered 19 December 2017.

## 1. Introduction

Clinical studies have demonstrated that pulmonary rehabilitation (PR) can be beneficial for lung mechanics (LM), chest kinematics (CS), respiratory and peripheral muscle function, cardiovascular response, metabolism (change in lactate threshold), exercise capacity, and quality of life [1]. Pulmonary rehabilitation programs need to include breathing training, controlled breathing techniques, endurance training, strengthening of respiratory and peripheral muscles, and nutritional and psychosocial support [1]. We can improve the exercise capacity, dyspnea, and quality of life of chronic obstructive pulmonary disease (COPD) patients with this complex program.

Dynamic hyperinflation produces a negative effect on the position and movement of the diaphragm in COPD, which has consequences for lung mechanics and chest kinematics. This phenomenon results in an increment in the end-expiratory lung volume and residual volume and leads to a reduction in the strength and endurance of the diaphragm [2,3,4]. The diaphragm is fixed in the inhalation position because of the decrease in expiratory flow, and its relaxation and range of motion are limited. As a result, diaphragm strength is reduced, resulting in increased dyspnea and reduced exercise capacity [3,4]. During dynamic hyperinflation (DH), the rib cage loses elasticity, and as a result, poor functioning of the respiratory muscles, atrophy, diminished spontaneous breathing, and dyspnea can develop [3,4]. The movement of the chest wall is associated with the movement of the diaphragm, and it is, therefore, important to strengthen the diaphragm in COPD. During breath holding, the muscles, especially the diaphragm and intercostal muscles, which are involved in breathing, move in a larger range; they are more relaxed and become stronger [1].

Optimization of the available lung volume is needed in COPD. Slowing down the respiratory rate by controlling the central nervous system is important for effective breathing. Our new freediving training technique is capable of optimizing the effectiveness of breathing [1].

Our new technique to reduce respiratory rate as a part of pulmonary rehabilitation in COPD came from the application of freediving training techniques in several sports in Hungary. We used the technique in national team sportsmen, who had advantages in breath-holding time and exercise capacity in competitions [5]. On the basis of these physiological effects, we wondered whether the respiratory techniques used by athletes could also be beneficial in the rehabilitation of COPD patients. When practicing air retention techniques used in the freediving training (FD) program, sportsmen and patients increase the air retention interval within their comfort zone, enabling longer breath holds and thus a reduced respiratory rate. A 15 s air retention time corresponds to four breaths per minute [5].

This new technique can help control respiration and optimize the effectiveness of breathing in terms of tidal volume.

We sought to determine the effectiveness of FD + COPD rehabilitation. There is a question about the favorable effect of freediving training in terms of breathing effectiveness: is the FD technique an advantageous additional treatment in conditions in which increasing breathing and respiratory effectiveness is recommended?

## 2. Patients and Methods

Forty-six patients with COPD (25 men and 21 women; median age 66 years; FEV_1_: 43 pred; BMI: 27 kg/m^2^) were assigned to the conventional PR program, and twenty-three COPD patients (15 men and 8 women; median age 63 years; FEV_1_: 41% pred; BMI: 28 kg/m^2^) were assigned to the FD + PR group. We considered FEV1 < 60%, FEV1/FVC < 70% with the maximal dose of bronchodilators, and age > 50 years as inclusion criteria; severe joint disease, NYHA III–IV heart failure, and acute exacerbation of COPD were the exclusion criteria. The algorithm for inclusion is shown in Figure 1. The inpatient PR program included chest wall mobilization, controlled breathing techniques, and personalized training programs for a bicycle and a treadmill in the PR group for 6 weeks [1] and 3 weeks of the PR program + 3 weeks of PR + FD in the PR + FD group. The patients in the FD group performed comfort zone breath-hold training for 30 min daily. For this technique, patients performed repeated breath-holding techniques for 30 min to reduce their respiratory rate. After a deep inhalation, they performed a breath hold until an elongated exhalation. They repeated these breath holds for 30 min, trying to hold their breath for longer periods. As an example, if they performed 15 s respiratory cycles (inhalation–breath-hold–exhalation), it would result in 4 breaths per minute. These patients performed a complex pulmonary rehabilitation program comprising chest wall stretching, physiotherapy, controlled breathing techniques, and a personalized exercise schedule two to three times per day consisting of 20–30 min of cycling and treadmill exercise for 3 weeks [1,6]. The personalized exercises were adjusted according to the patients’ BORG scores (dyspnea and leg fatigue) and their individual requirements. Changes in duration and intensity of the training were based on maintaining a BORG symptom score of 7 [1,6]. The protocol was defined considering the stage of COPD, the actual status of the heart and other comorbidities, blood gas value, and history of exacerbations [1].

Training programs were performed at a high intensity, either continuously or with interval modification. The training intensity was set at 80% of the peak work rate in the high-intensity continuous training group. The intensity was changed from 50 to 90% in the interval training group. Patients with pulmonary hypertension (PH) were involved in the interval training group [6].

Patients performed breathing exercises and training in the open-air corridor of our department, which has a special microclimate as an advantage. Chest physiotherapy included controlled breathing and stretching techniques, strength training, and chest and spine mobilization [1].

Breath-hold practices are currently used as a complement to conventional hospital rehabilitation. Patients increase their breath hold within their comfort zone in this type of maneuver. Exercises and breath-holding intervals are recorded with a self-developed breathing monitor belt. The exercise lasts half an hour daily. Before and after the three-week period, a health assessment is performed.

We collected the following functional parameters: 6 min walking distance (6MWD), lung function, chest wall expansion (CWE), grip strength, maximal inspiratory pressure (MIP), and breath-holding time (BHT). Quality of life, dyspnea, and the severity of the disease were monitored by the CAT, the mMRC, the BODE index, and an alternative scale developed by our group [7,8,9,10,11,12,13,14,15,16,17,18,19].

## 3. Measurements

### 3.1. Pulmonary Function

Lung function parameters were measured according to ATS/ERS guidelines and included dynamic lung function values (FEV1; FVC < FEV1/FVC) and resting slow vital capacity (VC) as a marker of resting hyperinflation of the chest [11]. We used the Piston lung function device (Piston, Budapest, Hungary) device for spirometry.

#### 3.1.1. Six-Minute Walking Distance

The 6MWD test was performed in a 30-meter-long corridor following the ATS/ERS guidelines [11]. Oxygen saturation, heart rate, and a modified Borg scale were evaluated before, during, and after the test. Patients were asked to walk as fast as possible, and the measurements were taken during the test [13].

#### 3.1.2. Chest Wall Expansion

We measured the circumference of the chest at the xiphoid process for deep inspiration and expiration. CWE was calculated as the difference between the two measurements. The result was based on three measurements, and the average was calculated on the basis of these values [14].

#### 3.1.3. Maximal Inspiratory Pressure

We measured the MIP with the PowerBreathe K1 device (POWERbreathe International Limited, Southam, UK). The measured values were related to the patient’s height, weight, age, and gender. Results were classified as “very poor,” “poor,” “average,” “fair,” “good,” and “very good”. The measurement was performed as a sudden inhalation with maximal force after a maximal exhalation. The average of three measurements was calculated [15].

#### 3.1.4. Breath-Holding Time

After a maximal inspiration, the patient was asked to hold the air in their lungs as long as possible. The nose was clipped and the mouth was shut during this procedure. Peripheral neural afferents from the respiratory system are activated by breath-holding and dyspnea, which may be related to the neural circuit responsible for processing motor output [16,17].

#### 3.1.5. Grip Strength Measurement

A Kern handgrip dynamometer (2016 Kern and Sohn GmbH, Balingen, Germany) was used to determine peripheral muscle force. Measurements were repeated three times, and the average of the three values was calculated [18,19].

#### 3.1.6. Quality of Life and DYSPNEA Questionnaires

Health status questionnaire and dyspnea score (CAT, mMRC, BODE index, and alternative scale): health status was evaluated using the CAT marker test, and dyspnea was screened using mMRC scoring [8,20]. The tests were conducted both before and after rehabilitation. The mMRC (modified Medical Research Council) dyspnea scale was developed by the British Respiratory Society.

### 3.2. Severity of the Disease

#### 3.2.1. BODE Index

The BODE index was calculated on the basis of BMI, FEV_1_, 6MWD, and mMRC [9].

#### 3.2.2. Alternative Scale

A new alternative scale was developed by our group. Similar to the BODE index, the new alternative scale was based on BMI, FEV_1_, and 6MWD. The mMRC scoring utilized by the BODE index was replaced by the CAT questionnaire (CAT 0–10 = mMRC score of 1, 11–20 = mMRC score of 2, 21–30 = mMRC score of 3, 31–40 = mMRC score of 4) in our new alternative scale [1,10].

### 3.3. Statistical Methods

Our first step in analyzing the data was to check the applicability of the parametric statistical tests. The vast majority of continuous variables did not follow a normal distribution (which was tested with the Saphiro–Wilk test); therefore, non-parametric statistical tests were performed. Wilcoxon’s signed rank test and Mann–Whitney U tests were used to compare the median values of the groups. Fisher’s exact tests were used to analyze the differences in categorical variables. Continuous data are presented as median values and interquartile ranges (IQR) and with box plots; categorical data are presented as sample sizes and proportions.

## 4. Results

We compared the two rehabilitation groups, one with and the other without the freediving technique. The conventional pulmonary rehabilitation program (PR) lasted 6 weeks, and the PR + FD program included a 3-week conventional rehabilitation and then PR + FD training. Table 1 contains the demographics and comorbidities of the two groups.

There were no significant differences between the two groups’ median age (*p* = 0.435) or gender distribution (*p* = 0.600). Furthermore, there were no significant differences in terms of the BMI (*p* = 0.529) or FEV1 (*p* = 0.862) values between the PR + FD and PR groups. The distribution of comorbidities, such as hypertension (*p* = 0.547), diabetes (*p* = 0.532), pulmonary hypertension (*p* = 0.608), and emphysema (*p* = 0.605), showed no statistical differences between the groups. The prevalence of respondents who had stopped smoking showed no statistical difference (*p* = 0.544) (Table 1).

The FEV1% pred was 40.5 (IQR = 19) before the rehabilitation, which significantly (*p* < 0.001) increased to 50 (IQR = 24) after rehabilitation in the group with the freediving technique. The same significant increase (*p* = 0.002) was observed in the group without the freediving technique, in which the FEV1% pred was 43 (IQR = 36) before rehabilitation and increased to 50 (IQR = 27).

No significant (*p* = 0.871) change was observed regarding in the group with the freediving technique; FVC% pred was 75 (IQR = 19) at the beginning of the program, and the median value was the same after rehabilitation. Only a minor decrease was observed in the interquartile range (IQR = 15). However, in the other group, a significant increase (*p* = 0.002) was observed, with the value increasing from 73 (IQR = 28) to 80 (IQR = 36).

The IVC significantly increased in both groups; its initial value was 2.51 (IQR = 1.01) L and it increased to 2.78 (IQR = 1.08) L in the freediving group (*p* < 0.001); in the other group, it increased from 2.23 (IQR = 1.32) L to 2.25 (IQR = 1.35) L (*p* = 0.011).

Before rehabilitation in the group with the conventional pulmonary program and the freediving technique, the chest expansion was 3 (IQR = 2.5) cm, and a significant increase (*p* < 0.001) was shown; it increased to 6.25 (IQR = 2.5) cm. In the other group, the same significant increase (*p* < 0.001) was observed; however, the increase was considered smaller compared with that of the other group because it was 3.5 (IQR = 3.5) cm before the program, and it increased to 5 (IQR = 4) cm after the rehabilitation.

Breath-holding time significantly increased in both groups. In the intervention group. it was 24 (IQR = 18) s which increased to 41 (IQR = 30) s (*p* < 0.001); in the conventional group, it was 20 (IQR = 10) s, and it increased to 24 (IQR = 16) s (*p* = 0.002).

A significant increase was observed in MIP, which was 65 (IQR = 27) before the rehabilitation program in the freediving group and significantly increased (*p* < 0.001) to 79 (IQR = 25). However, the same significant association (*p* < 0.001) was shown in the group with the conventional program, in which its value increased from 64 (IQR = 27) to 69 (IQR = 36).

Significant changes were observed in grip strength during the program. In the freediving group, it increased from 27.3 (IQR = 15.2) kg to 31.3 (IQR = 18.1) kg (*p* < 0.001); in the other group it increased from 27.2 (IQR = 12) kg to 27.8 (IQR = 13.5) kg (*p* < 0.001).

A significant decrease was observed in both groups regarding CAT; in the freediving group, a statistically significant decrease (*p* < 0.001) was shown, with the value reducing from 17 (IQR = 9) to 7 (IQR = 7). In the other group, which received the conventional pulmonary rehabilitation program only, the decrease was significant (*p* < 0.001) as well, with the value decreasing from 18 (IQR = 14) to 12 (IQR = 13).

Significant (*p* = 0.002 and *p* = 0.011) changes were observed related to mMRC in both groups; however, the median values did not change. This means that there were significant changes regarding mMRC; however, the changes could only be seen on an individual level, not when the cases were pooled together.

The alternative scale score significantly decreased in both groups; in the freediving group, it decreased from 6 (IQR = 2) to 2 (IQR = 2) (*p* < 0.001), and it decreased from 4 (IQR = 3) to 3 (IQR = 3) in the conventional group (*p* < 0.001).

The 6MWD increased in both groups. In the group with the extra freediving program, the 6MWD before the rehabilitation was 399.55 (IQR = 116) m, and it increased to 490 (IQR = 96) m (*p* < 0.001). An increase was seen in the other group, in which the 6MWD was 345 (IQR = 99) m before the rehabilitation program, and it increased to 382 (IQR = 86) m after the rehabilitation (*p* < 0.001).

We found significant differences regarding FVC% pred, chest expansion, breath-holding time, MIP, CAT, mMRC, alternative scale, and 6MWD when comparing the differences before and after rehabilitation in the two groups. However, the differences related to FEV1% pred, IVC, and grip strength were not significant. We detected a significant improvement in the PR + FD group regarding these parameters (Table 2) as well. The improvements were significantly higher in the PR + FD group than in the PR group in 6MWD, CWE, MIP, BHT, CAT, mMRC, FVC, and the alternative scale. Lung function in terms of FVC% pred did not change significantly in either group in all cases.

Rehabilitation produced a favorable effect on lung mechanics, chest kinematics, respiratory and peripheral muscle function, exercise capacity, and quality of life (Table 2) in the conventional rehabilitation group.

We detected a significantly higher 6MWD in the PR + FD group compared with the PR group (Figure 2). A significantly higher improvement in disease severity was measured in the BODE index in the PR + FD group compared to the PR group (Figure 3).

## 5. Discussion

We conducted a clinical study on COPD patients in a 6-week conventional PR program compared with a 3-week PR + 3-week PR + FD program. The conventional PR program included chest wall mobilization, controlled breathing techniques, and personalized training programs for the bicycle and treadmill, and in the FD program, patients increased their breath hold within their comfort zone for 30 min. The PR + FD program resulted in a significantly larger improvement in maximal exercise capacity, chest kinematics, muscle strength, and quality of life monitored by the CAT compared with the PR program alone. 

The use of the freediving training technique in the rehabilitation of COPD patients had a positive effect on respiratory physiology, respiratory mechanics, and chest kinematics. The technique aimed to improve respiratory efficiency both anatomically and physiologically. Divers achieve outstanding results by optimizing their breathing. These training methods result in the improvement of the breathing efficiency of patients with breathing difficulties in COPD also. From an anatomical point of view, the method involves the use of lower abdominal breathing, which results in increased activity of the diaphragm and the respiratory muscles involved in breathing, increasing the strength of the diaphragm and intervertebral muscles and reducing the constriction of ventilation. By pushing the lower abdomen, the internal organs and the compartment move downward, increasing the proportion of the thoracic cavity so that the inhaled air reaches the lower alveolar-rich regions of the lungs. There is more time for gas exchange as a result of slow breathing. The retention expands the airways, allowing the alveoli to drain properly during exhalation. We could observe all these positive physiological effects of freediving training, which manifested directly in the improvement of breath-holding capability and indirectly in chest kinematics, maximal exercise capacity, muscle strength, and quality of life.

Considering the effects of the freediving technique, we need to focus on the physiology of breathing and the respiratory system. The respiratory rate and tidal volume define minute ventilation. Tidal volume is divided into two components, and one of them is dead space. This part of the volume does not contact the functioning pulmonary capillaries during ventilation [21,22,23]. Because of the absence of air–blood exchange, O_2_ cannot communicate with the circulation, and CO_2_ cannot be removed in this part. The other portion of the tidal volume is called alveolar volume [21]. The air can be in contact with the functioning capillaries in this part. Air spaces are terminal respiratory units and include respiratory bronchioles, alveolar ducts, alveolar sacs, and alveoli in this portion. The alveolar volume component of each tidal breath is effective for gas exchange as a result of freediving training. Minute ventilation is increased by the tidal volume increment, which has a greater effect on gas exchange [21]. 

We technically divide the respiratory system into three regions—the lower abdomen, rib cage, and lung tip—in freediving training, The lower abdomen is not part of the respiratory system, but it is one of the three regions of the body involved in and assisting the respiratory movements, along with the rib cage and the lung lobes. Inhalation and exhalation also start from the lower abdomen. The belly is squeezed during inhalation as if sucking air into the navel [21]. During exhalation, these regions are cleared in the same order, so the lower abdomen is the focal point of respiration and is where the exhalation and inhalation begin [21]. Exhalation must also be complete. Using the lower abdomen, rib cage, and lung tip in breathing and lower abdominal breathing increases the alveolar volume with each inhalation, i.e., the rate of dead space decreases, thereby increasing respiratory efficiency [21]. We indirectly observed better reactions in the thoracic compartment while monitoring the improvement in chest kinematics. 

Throughout the PR + FD group breathing exercise, efforts should be made to increase the volume and breath-holding interval within the comfort zone (Figure 4). We can achieve a lower respiratory rate and more effective breathing with longer intervals and greater volume. It is important to perform these maneuvers within the comfort zone. This exercise is based on the techniques used in freediving training, which we used in the PR + FD group.

Ventilation at rest requires only the inspiratory muscles in most individuals. Expiration is usually passive and is secondary to the respiratory system returning to its resting state. Therefore, the inspiratory time is the period of active respiratory pacemaker output with quiet breathing [22,23]. Adjusting the rate, length, and intensity of neural output from the pacemaker leads to changes in the number of breaths per minute and the volume of each inspiration or tidal volume [21]. These final outputs of the respiratory pacemaker, the rate and tidal volume, are the two components of ventilation. The expiratory muscles begin to play a role in cases of disease or increased ventilatory demands [21]. When this occurs, the length of time required to empty the lungs adequately also leads to changes in the respiratory rate and tidal volume. In the PR + FD training, we optimized the tidal volume and respiratory rate relationship. 

### Elements of Conventional Rehabilitation

Individual endurance training includes exercise on a stationary bicycle/treadmill and a manual treadmill. It is applied in continuous and interval form [6,24,25,26,27,28,29,30,31,32]. Interval training involves one minute of work followed by one minute of rest. The intensity is based on the BORG scale and the Visual Analog Scale (VAS) [5]. All training programs were individualized for each patient; we used interval training for patients who had pulmonary hypertension. 

Breathing and stretching exercises, controlled breathing exercises, and PLB (Pursed Lip Breathing) can cause increased pressure in the respiratory system during exhalation and lead to small airway dilatation, increases in expiratory flow, and decreases in chest hyperinflation [1]. Spine and chest mobilization can increase the range of motion of the rib cage. It can improve the ventilation of multiple lung areas. Controlled breathing exercises can make breathing and exhalation more effective and increase diaphragm contraction and relaxation [1,21]. These effects are further enhanced, thereby strengthening the diaphragm and increasing the range of motion of the ribcage during comfort zone breath-hold training (CBT). Restraints also increase the diameter of the small airways, reducing DH. Before performing the exercise, we draw attention to the importance of lower abdominal breathing. In this case, the conscious focus of breathing is placed in the lower abdomen. By moving the lower abdomen forward and downward, the deepest inhalation is achieved, thus helping to ventilate all areas of the lungs. The diaphragm pulls the lungs downward, pushing the internal organs down, increasing the proportion of the thoracic cavity, and delivering the inhaled air to the lowest alveolar-rich sections of the lungs so that gas is exchanged throughout the lungs. Following the inhalation of the abdomen, the rib cage is opened, and in the last phase of the inhalation, the air reaches the upper lung apex. In addition to anatomical effects, CBT also contributes to better breathing by optimizing physiological processes. During slow breathing, it takes more time for the gases to change, so breathing takes place not only on a larger surface but also in a longer time interval during each breath. In this form of breathing, the oxygen utilization of the respiratory muscles is also more optimal in relation to the volume of respiration because the work of the respiratory muscles is proportionally smaller, and their oxygen utilization is greater [1]. We observed that all indirect physiological effects were positive with the increase in BHT. According to our hypothesis, freediving training techniques can improve the status of COPD patients more than conventional rehabilitation in terms of respiratory physiology and respiratory functional parameters [21].

The application of freediving training methods is considered a novelty in the health system. Freedivers have unique experience in developing breathing efficiency. Incorporating these experiences in the treatment of diseases associated with a decrease in respiratory efficiency represents a new direction in pulmonary rehabilitation. It is safe for the patients if they are using this technique within their comfort zone.

## 6. Limitations of the Study

The study was conducted in an inpatient program with one 30 min CBT session per day. Increases in the number of patients and an outpatient rehabilitation study are needed. During this phase, patients would use a respiratory monitor at home for one month (2 × 30 min daily). We would perform a complex assessment before and after rehabilitation. The breathing monitor device comes with a downloadable application for smartphones, which includes breathing training and transmits and records exercise data and breathing patterns in real time, which provides information about the length of inhalation, exhalation, and air retention. The recorded data show short- and long-term adaptation.

In conclusion, PR + FD training resulted in longer BHT and had an additive positive effect on exercise capacity, chest kinematics, muscle strength, and quality of life. PR + FD was as safe as PR training, and there were no side effects during this type of training.

## 7. Conclusions

As the result of our study we found, that PR + FD program can be more effective than PR program alone in the rehabilitation of COPD patients. Significantly larger improvement was measured in maximal exercise capacity, chest kinematics, muscle strength and quality of life monitored by the CAT compared with the PR program.

## Figures and Tables

**Figure 1 ijerph-19-11549-f001:**
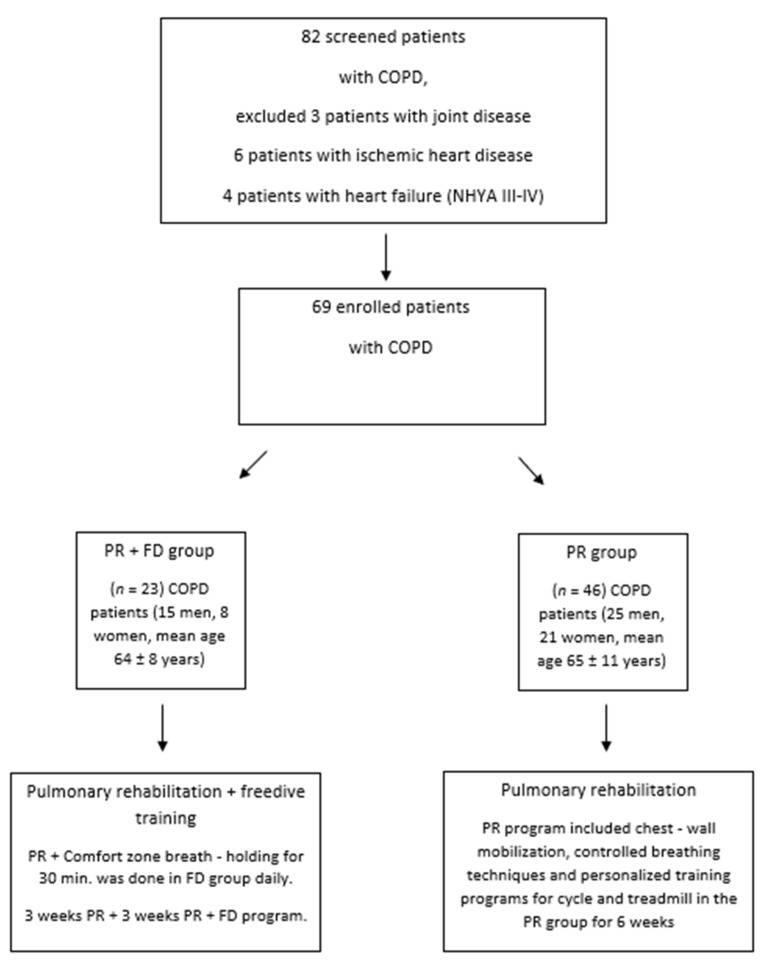
Algorithm of the study.

**Figure 2 ijerph-19-11549-f002:**
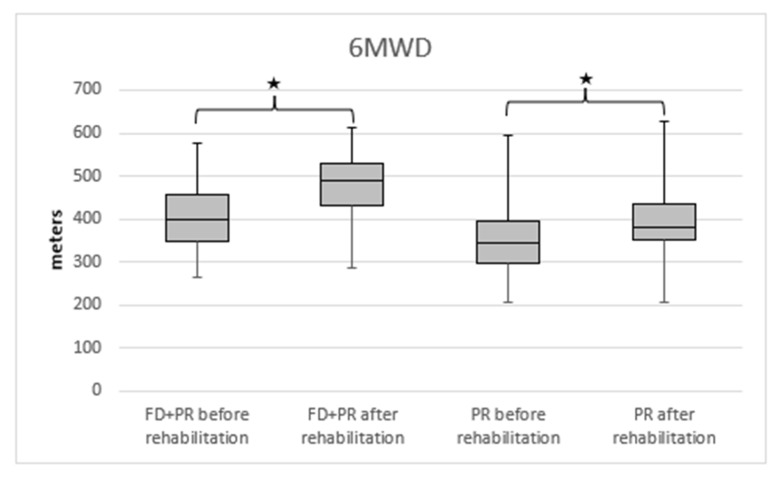
The change in 6MWD in PR and PR + FD groups in COPD; 6MWD: 6 min walking distance, PR: pulmonary rehabilitation, FD: freediving training, * *p* < 0.001 after vs. before rehabilitation.

**Figure 3 ijerph-19-11549-f003:**
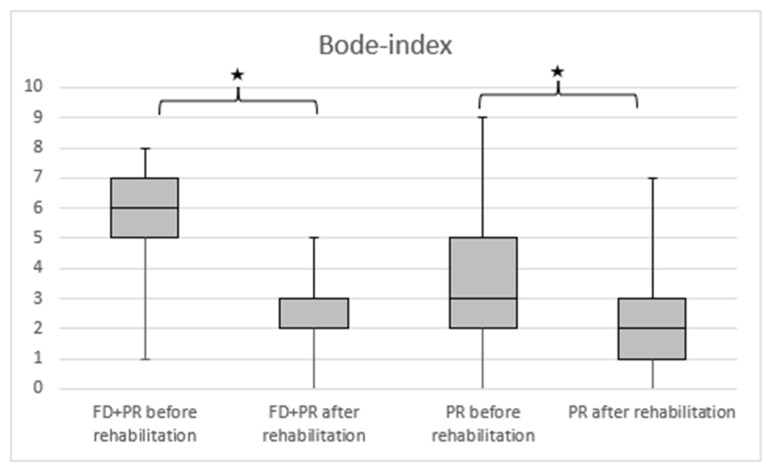
The change in the severity of COPD (measured by BODE index) in the PR and PR + FD groups. BODE index: FEV_1_, 6MWD, BMI, and mMRC; PR: pulmonary rehabilitation; FD: freediving training; * *p* < 0.001 after vs. before rehabilitation.

**Figure 4 ijerph-19-11549-f004:**
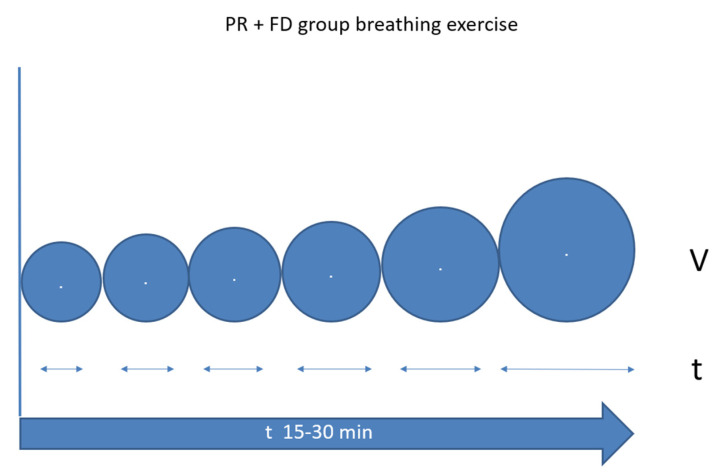
Schematic figure of breath-hold time and volume change during PR + FD rehabilitation. FD: freediving training, PR: pulmonary rehabilitation.

**Table 1 ijerph-19-11549-t001:** Characteristics of the two groups.

Characteristics	Group PR (*n* = 46)	Group PR + FD (*n* = 23)	*p*-Value
Age (years)	66 ± 11	63 ± 12	0.435
Male/Female	24/22	12/11	0.600
BMI (kg/m^2^)	27 ± 9	28 ± 8	0.529
FEV_1_ (%pred)	43 ± 36	41 ± 19	0.862
Hypertension	35 (76%)	18 (78%)	0.547
Diabetes	11 (24%)	6 (26%)	0.532
Pulmonary hypertension	12 (26%)	6 (26%)	0.608
Emphysema	14 (30%)	7 (30%)	0.605
Quitting of smoking	33 (72%)	17 (74%)	0.544

BMI: body mass index, FD: freediving training, FEV_1_: forced expiratory volume in the first second, PR: pulmonary rehabilitation.

**Table 2 ijerph-19-11549-t002:** Change of functional markers in groups 1 and 2.

	PR Group Before Rehabilitation	PR Group	*p*-Value	PR + FD Group	PR + FD Group	*p*-Value	Comparison of PR + FD and PR *p*-Value
		After Rehabilitation		Before Rehabilitation	After Rehabilitation		
FEV_1_ % pred	43 (36)	50 (27)	0.002	40.5 (19)	50 (24)	<0.001	0.101
FVC % pred	73 (28)	80 (36)	0.002	75 (19)	75 (15)	0.871	0.035
IVC (L)	2.23 (1.32)	2.25 (1.35)	0.011	2.51 (1.01)	2.78 (1.08)	<0.001	0.127
Chest expansion (cm)	3.5 (3.5)	5 (4)	<0.001	3 (2.5)	6.25 (2.5)	<0.001	<0.001
Breath-holding time (s)	20 (10)	24 (16)	0.002	24 (18)	41 (30)	<0.001	<0.001
MIP (H_2_Ocm)	64 (27)	69 (36)	<0.001	65 (27)	79 (25)	<0.001	0.009
Grip strength (kg)	27.2 (12)	27.8 (13.5)	<0.001	27.3 (15.2)	31.3 (18.1)	<0.001	0.059
CAT	18 (14)	12 (13)	<0.001	17 (8)	7 (7)	<0.001	0.006
mMRC	2 (0)	2 (1)	0.011	2 (1)	2 (1)	0.002	0.010
Alternative scale	4 (3)	3 (3)	<0.001	6 (2)	2 (2)	<0.001	<0.001
6MWD (m)	345 (99)	382 (86)	<0.001	399.5 (116)	490 (96)	<0.001	<0.001

FD: freediving training, FEV_1_: forced expiratory volume in the first second, FVC: forced vital capacity, IVC: inspiratory vital capacity, MIP: maximal inspiratory pressure, CAT: COPD Assessment Test, mMRC: modified Medical Research Dyspnea Scale, alternative scale (FEV_1_, BMI, 6MWD, CAT), PR: pulmonary rehabilitation, 6MWD: six-minute walking distance.

## Data Availability

The dataset supporting the conclusions of this article is available from the corresponding author onuponeasonable request.

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
