# Peer review of "Freedive Training Gives Additional Physiological Effect Compared to Pulmonary Rehabilitation in COPD"

_ijerph, 2022, doi:10.3390/ijerph191811549_

Round 1

Reviewer 1 Report

Here, Csizmadia Z et al. investigate the efficacy of 3 weeks freedive training after 3 weeks of conventional pulmonary rehabilitation (PR+FD) in COPD patients compared to 6 weeks of pulmonary rehabilitation (PR). Authors show that PR+FD improves the pulmonary capacity further than only PR. The implementation of PR+FD for COPD patients can help the daily tasks of around 16 million patients only in the US.

Minor concerns: 

- COPD can arise due to different reasons, emphysema or bronquitis. Has the etiology of COPD been taking into account for this study? Would these affect the outcome of the training?

- Reference #8 cannot be found. Authors should make sure any reference is accessible to the public.

- Fix font for references to make them consistent.

Author Response

Dear Reviewer,

Thank you very much for your time and work with our manuscript. We would like to reply with the following to the reviewers` questions and comments.

Answers for the reviewer`s questions:

  1. COPD can arise due to different reasons, emphysema or bronquitis. Has the etiology of COPD been taking into account for this study? Would these affect the outcome of the training?

Thank you for your comment, this research group was homogeneous on both groups, 30% of the patients had an emphysema + COPD and resting hyperinflation based on body pletysmography. Chest hyperinflation manifested in this group and this type of method with reduced respiratory rate can help to reduce the chest hyperinflation.

  1. Reference #8 cannot be found. Authors should make sure any reference is accessible to the public.

Thank you for your comment, we deleted the reference No 8.

  1. Fix font for references to make them consistent.

Thank you for your comment, we have fixed the font of the references.

Hoping that the revised version will meet all the requirements of your journal, we thank you again for your efforts to improve the manuscript and look forward to your favorable news.

Yours Sincerely

Zoltán Csizmadia

Reviewer 2 Report

The authors have compared two lines of treatments for improving lung functioning after COPD diagnosis and have shown data that determines Freedive Training + Pulmonary rehabilitation (PR) has better outcomes compared to PR alone. The study is performed comprehensively and several parameters are tested. However, the authors need to address a few issues in the paper:

1. It would be helpful to the readers if the authors described some of the parameters tested in more detail such as FEV1 and 6MWD etc.

2. Do the authors expect better comes if six weeks of FD + PR training were done on patients? Or initial PR training is necessary for a better outcome.

3. Sample size for patients in both groups is different and could that attribute to some change observed in both groups

Author Response

Dear Reviewer,

Thank you very much for your time and work with our manuscript. We would like to reply with the following to the reviewers` questions and comments.

Answers for the reviewer`s questions:

  1. It would be helpful to the readers if the authors described some of the parameters tested in more detail such as FEV1 and 6MWD etc.

Thank you for your comment, we added FEV1 and 6MWD to the table.

  1. Do the authors expect better comes if six weeks of FD + PR training were done on patients? Or initial PR training is necessary for a better outcome.

Thank you for your comment, it is true, that the patients performed classical pulmonary rehabilitation (PR) for the first three weeks in this study. If we are using PR+FD from the onset of PR has additive effect, also. So it has a significantly different effect of PR+FD to PR if we start the program either with PR+FD or in a later phase.

  1. Sample size for patients in both groups is different and could that attribute to some change observed in both groups.

Thank you for your comment, it is known, that sometimes there is a 1:2 ratio in clinical studies and we used a similar sample for the analysis. The lower number of the patients with a statistical difference even show the effectiveness of the PR+FD to PR training

Hoping that the revised version will meet all the requirements of your journal, we thank you again for your efforts to improve the manuscript and look forward to your favourable news.

Yours Sincerely

Zoltán Csizmadia

Reviewer 3 Report

The study aims at demonstrating the benefits of freedive training (FD) method, a techniques used by athletes, when added to standard pulmonary rehabilitation (PR) in COPD patients. As a matter of fact, it can be postulated that patients undergoing FD increase the air ritention interval within the comfort zone, so that longer breath-holds means reduced respiratory rate. This can help them control respiration so as to optimize the effectiveness of breathing in terms of tidal volume. 

The authors took into accounts several parameters, such 6mwt, FEV1, FVC, chest wall expansion and breath holding time in each group, before and after PR. The authors conclude that PR + FD improved chest kinematics better than the sole PR. 

However interesting, the study needs several improvements (see specific comments) and lacks the clinical significance it aimed at. Therefore, it needs major revision. 

Specific comments

Title: too generic, it should describe a little better what the authors found 

Abstract: some passages are not as clear as they should be (such as the one about breath holding). Please improve it. 

Introduction: The paragraphs on PR and DH could be improved, quoting more papers and describing also other pathological mechanisms of DH, not only the diaphragm impairment.  it is not clear if the technique introduced in your PR+FD group is different from the one previously used on athletes. Moreover, since the quotes refer to oral communications from a conference, it is not clear whether it was the authors of the present paper who came up with such technique. Please modify the section as above stated. 

Patients and methods: How many patients were screened for inclusion? Were they consecutively enrolled? How was the "severe joint injury" assessed? In which consists your special microclimate? 

Measurements: which kind spirometer was used to assess lung function? Regarding MIP: how were the measures classified qualitatively? Breath-holding time: the passage regarding neural circuits should not be included in this section. 

Results: Table 2 shows the changes of functional markers in both groups. Several significative differences are shown, but it lacks 6MWD and BODE Index, which have been plotted into two different graphics. It is stated that several markers have a statistical significative difference between the two groups, but it is not clear whether the basal data have or not a statistically significative difference. Between groups comparisons have not been collected in a table, and this makes it difficult to double check the data. 

Discussion: here the authors should describe the current state of the art and enlighten the novelty of their findings, as well as their clinical significance and their clinical applications, aspects which should be improved. 

Language: some minor revisions are needed. 

Author Response

Dear Reviewer,

Thank you very much for your time and work with our manuscript. We would like to reply with the following to the reviewers` questions and comments.

Answers for the reviewer`s questions:

  1. Title: too generic, it should describe a little better what the authors found 

Thank you for your comment, we agree with your opinion and we change the title to „Freedive training gives additional physiologic effect compared to pulmonary rehabilitation in COPD”.

  1. Abstract: some passages are not as clear as they should be (such as the one about breath holding). Please improve it. 

Thank you for your comment, we corrected it. We described the breath holding time in the „Methods” session as no. 3.1.4 in more details.

  1. Introduction: The paragraphs on PR and DH could be improved, quoting more papers and describing also other pathological mechanisms of DH, not only the diaphragm impairment.  it is not clear if the technique introduced in your PR+FD group is different from the one previously used on athletes. Moreover, since the quotes refer to oral communications from a conference, it is not clear whether it was the authors of the present paper who came up with such technique. Please modify the section as above stated. 

Thank you for your comment, I confirm that the FD technique which we used was the same at the athletes and COPD patients, we developed it first for athletes. We deleted that reference papers from the reference list, because they were not manuscripts.

  1. Patients and methods: How many patients were screened for inclusion?

Thank you for your comment, we screened 102 patients and 69 were eligible for the study because of the inclusion criteria.  

Were they consecutively enrolled?

Thank you for your comment, yes, they were consecutively enrolled.

How was the "severe joint injury" assessed?

Thank you for your comment, we used physical examinations for severe joint injury. We considered the joint injury anamnesis of the patients, and we were looking of the movement of the joints which is affected to the PR problems.

In which consists your special microclimate?

Thank you for your comment, the special climate had positive effect of the bronchial hyper reactivity of the patients, but it does not matter in our new method.

  1. Measurements: which kind spirometer was used to assess lung function? Regarding MIP: how were the measures classified qualitatively? Breath-holding time: the passage regarding neural circuits should not be included in this section. 

Thank you for your comment, Piston spirometer (Piston, Budapest, Hungary) was used to asses lung function, we evaluated the MIP with the Power Breathe K1 (POWERbreathe International Limited, Southam, UK).  We agree and we made the modifications at the measurement.

  1. Results: Table 2 shows the changes of functional markers in both groups. Several significative differences are shown, but it lacks 6MWD and BODE Index, which have been plotted into two different graphics. It is stated that several markers have a statistical significative difference between the two groups, but it is not clear whether the basal data have or not a statistically significative difference.

Thank you for your comment, we confirm, that there was no significant difference in any functional parameters between the two groups. We do not give the 6MWD and BODE index again in the table, because it can be a double writing of the data.   

Between groups comparisons have not been collected in a table, and this makes it difficult to double check the data. 

Thank you for your comment, we made a table with the two groups separately but when we see the p value separately we can see the significant intergroups difference (for example FD+PR p< 0.01 vs. PR p<0.05)

  1. Discussion: here the authors should describe the current state of the art and enlighten the novelty of their findings, as well as their clinical significance and their clinical applications, aspects which should be improved. 

Thank you for your comment, We incorporated a section in the discussion in terms of the novelty of freediving training as well as clinical significance and clinical application..  

  1. Language: some minor revisions are needed.

Thank you for your comment, we asked a native English language speaker to correct the manuscript

Hoping that the revised version will meet all the requirements of your journal, we thank you again for your efforts to improve the manuscript and look forward to your favourable news.

Yours Sincerely

Zoltán Csizmadia

Round 2

Reviewer 3 Report

There have been improvements in comparison with the first version of the manuscript. However, the language needs further improvement as well as the presentation of data and results. 

Author Response

Dear Reviewer,

Thank you very much for your time and work with our manuscript. We would like to reply with the following to the reviewers` questions and comments.

Answers for the reviewer`s questions:

There have been improvements in comparison with the first version of the manuscript. However, the language needs further improvement as well as the presentation of data and results. 

­Answer: We improved the language of the manuscript, restructured figures and tables and added new statistical data.

With kind regards,

Zoltán Csizmadia